# Identification of a Novel Gene Signature with DDR and EMT Difunctionalities for Predicting Prognosis, Immune Activity, and Drug Response in Breast Cancer

**DOI:** 10.3390/ijerph20021221

**Published:** 2023-01-10

**Authors:** Pan Zhang, Quan Li, Yuni Zhang, Qianqian Wang, Junfang Yan, Aihua Shen, Burong Hu

**Affiliations:** 1Department of Radiation Medicine, School of Public Health and Management, Wenzhou Medical University, Wenzhou 325035, China; 2Key Laboratory of Watershed Science and Health of Zhejiang Province, School of Public Health and Management, Wenzhou Medical University, Wenzhou 325035, China

**Keywords:** breast cancer, gene signature, cancer prognostic model, DNA damage repair (DDR), epithelial–mesenchymal transition (EMT)

## Abstract

Breast cancer, with an overall poor clinical prognosis, is one of the most heterogeneous cancers. DNA damage repair (DDR) and epithelial–mesenchymal transition (EMT) have been identified to be associated with cancer’s progression. Our study aimed to explore whether genes with both functions play a more crucial role in the prognosis, immune, and therapy response of breast cancer patients. Based on the Cancer Genome Atlas (TCGA) cancer database, we used LASSO regression analysis to identify the six prognostic-related genes with both DDR and EMT functions, including *TP63*, *YWHAZ*, *BRCA1*, *CCND2*, *YWHAG*, and *HIPK2*. Based on the six genes, we defined the risk scores of the patients and reasonably analyzed the overall survival rate between the patients with the different risk scores. We found that overall survival in higher-risk-score patients was lower than in lower-risk-score patients. Subsequently, further GO and KEGG analyses for patients revealed that the levels of immune infiltration varied for patients with high and low risk scores, and the high-risk-score patients had lower immune infiltration’s levels and were insensitive to treatment with chemotherapeutic agents. Furthermore, the Gene Expression Omnibus (GEO) database validated our findings. Our data suggest that *TP63*, *YWHAZ*, *BRCA1*, *CCND2*, *YWHAG*, and *HIPK2* can be potential genetic markers of prognostic assessment, immune infiltration and chemotherapeutic drug sensitivity in breast cancer patients.

## 1. Introduction

Breast cancer (BRCA) is one of the most prevalent malignant tumors endangering female health worldwide. According to the latest cancer statistics released by the International Agency for Research on Cancer (IARC), BRCA, surpassing lung cancer, had the top morbidity globally. Currently, the primary treatments for BRCA include surgery, radiotherapy, and hormonal therapy [1]. Despite significant improvements in BRCA treatment, the high morbidity and mortality remain major challenges for BRCA patients’ survival [2]. Recently, several researchers have found that methylation-related features [3], helicase-related markers [4], and immune-related markers [5] can be used as prognostic markers for BRCA patients. However, the prognosis of some patients may be complicated to predict precisely due to the heterogeneity [6] and metastasis of BRCA [7]. Therefore, it is of paramount importance to discover novel prognostic markers for implementing precise clinical therapy and prognosis evaluation.

DNA damage, induced by multiple toxic environmental stimuli, can be repaired by the cell’s internal DNA damage repair (DDR) pathways. Incorrect repair is one of the critical biological markers for cancer progression [8]. Numerous studies have illustrated that DDR serves as a barrier for tumorigenesis in the early stages of BRCA. Nevertheless, it can promote the malignant growth of tumor cells with defective genomic maintenance mechanisms [9]. Additionally, DDR is associated with radiotherapy prognosis for BRCA. The nuclear factor erythroid-2-related factor 2 (Nrf2) enhances the BRCA’s radiotherapy sensitivity by inhibiting DDR [10]. The tumor suppressor miR-139-5p may regulate radiation resistance in BRCA cells by suppressing DDR and ROS defense, promoting patients’ prognosis [11]. MMP14, a zinc-dependent matrix metalloproteinase, causes radiation resistance in triple-negative breast cancer (TNBC) by activating DDR, leading to a poor prognosis for patients [12]. As a result, DDR is dramatically related to the BRCA prognosis.

Epithelial–mesenchymal transition (EMT), a biological process for cancer development [13], plays an essential role in the early stage of cancer metastasis [14]. Studies have demonstrated that EMT is deemed as the crucial mechanism for cancer metastasis [15]. Meanwhile, EMT can modulate vascular permeability and subsequently promote cancer migration [16]. Moreover, there are also some associations between EMT and DDR. Peijing Zhang et al. revealed that in BRCA cells, ZEB1 (EMT-inducible transcription factor) binds to USP7 to stabilize CHK1, promoting DDR of the homologous recombination (HR) pathway [17]. RNF8 activates the nuclear localization of Twist (EMT-inducible transcription factor) to regulate DDR [18]. Furthermore, EMT is significantly relevant to BRCA prognosis. Cisplatin can suppress the growth of BRCA by blocking the rearrangement of the early EMT cytoskeleton, benefiting patients’ prognosis [19]. JAM2 restrains BRCA cell metastasis by inhibiting EMT-related pathways, improving patients’ outcome [20]. As we can see, EMT plays a vital role in the progression of BRCA.

In summary, DDR and EMT are closely related to the prognosis of BRCA patients. In addition, there are associations between DDR and EMT. At present, most research models for BRCA prognosis are based on single-function genes. However, due to the complexity of BRCA etiologies, the single-function genes’ prognosis model may not achieve the predictive effect expected. Therefore, we selected the dual-function (DDR and EMT) genes (here called DEDGs) and aimed to construct a model for predicting the prognosis of BRCA patients. Moreover, we further analyzed the differences in immune activity and chemotherapy sensitivity, providing potential guidance for targeted therapy and immunotherapy in the clinic for BRCA.

## 2. Materials and Methods

### 2.1. Data Collection

The transcriptome expression data and clinical information of 1072 primary breast cancer and 99 adjacent normal tissue samples were downloaded from the TCGA (https://portal.gdc.cancer.gov/) database, accessed on 28 July 2021. In addition, 2 independent validation cohorts, including GSE20685 (327 BRCA samples) and GSE88770 (117 BRCA samples), were obtained from the GEO database (https://www.ncbi.nlm.nih.gov/, accessed on 14 November 2022). All raw counts data were normalized by the R package “EDASeq” to be transformed by log2 (data + 1). TPM was transformed by log2 (data + 1) as the gene’s expression level. A total of 296 DDR genes were attained from GeneCards (https://www.genecards.org/), and 1011 epithelial–mesenchymal transition-related genes (EMT) were derived from dbEMT2 (http://www.dbemt.bioinfo-minzhao.org/) (Appendix A).

### 2.2. Differentially Expressed Gene Analysis

We used the R package “venn” and “VennDiagram” to take the intersection gene set of EMT and DDR, called DEDGs (Appendix A). Furthermore, the R package “DESeq2” was used to identify the differentially expressed genes (DEGs) between tumor and normal tissues with the criteria of |log2-fold change| >= 0.5 and *p*-value < 0.05 (Appendix A). Eventually, we extracted the intersection genes for DEDGs and DEGs. (Appendix A).

### 2.3. Unsupervised Consensus Clustering Analysis

Excluding patients without complete clinical information, 1050 BRCA patients were classified into different subtypes based on the DEDGs by an unsupervised consensus clustering method and comparing the survival analysis between the two subtypes. The unsupervised consensus clustering algorithm was performed by the “ConsensusClusterPlus” package, which has been widely used in cancer-related studies.

### 2.4. Construction and Validation of the Prognostic Model Based on DEDGs

The prognostic model was developed based on the TCGA-BRCA cohort (1050 samples, excluding patients without complete survival information). External BRCA cohorts (GSE20685 and GSE88770) were used to validate the predictive capability of the prognostic model.

The prognostic values for DEDGs were determined by univariate Cox regression analysis, where *p* < 0.05 was considered statistically significant. Subsequently, we used the least absolute shrinkage and selection operator (LASSO) regression to further screen candidate DEDGs for building a prognosis model using the R package “glmnet”. The prognostic risk score was determined by a linear combination of regression coefficients (β) and gene expression levels in the LASSO model. Risk score = $\sum_{i}^{k}$Xi × Yi (Xi: a gene Coefficient, Yi: a gene expression level). For example, the gene expression levels (Yi) of “*B*” and “*C*” in patient A are “B1” and “C1”, and the correlation coefficients (Xi) of these 2 genes are “B2” and “C2”. So, the patient A’s risk score = (B2 × B1) + (C2 × C1). Patients in the TCGA and GSE cohorts were classified into high- and low-risk groups based on the same median risk score.

The Kaplan–Meier (K-M) survival curve analysis was performed to compare overall survival (OS) between the two groups using the R package “survival”. Principal component analysis (PCA) based on the DEDGs was plotted by the “prcomp” function of the R package. To better evaluate the predictive performance of the DEDGs prognostic model, the receiver operator characteristic (ROC) curves for 5-, 7-, and 10-year survival were constructed using the R packages “timeROC” and “survival”.

### 2.5. Establishment of Prognosis Nomogram Combined with Clinical Features

The Univariate COX regression model was applied to determine prognostic features based on the risk score and clinical characteristics in TCGA-BRCA patients. Subsequently, variables with *p* < 0.05 were included in the multivariate Cox regression analysis to assess independent prognostic factors for BRCA. In the end, with the multivariate Cox regression analysis results, we employed the R package “rms” to create a nomogram to predict patients’ survival. To appraise the predictive capacity of the nomogram, the calibration curves for BRCA patients at 5, 7, and 10 years were established.

### 2.6. Functional Enrichment Analyses

We identified the DEGs between the high- and low-risk groups with a criterion of |log2-fold change (FC)| >= 0.8 and *p*-value < 0.01. The DEGs were annotated to Kyoto Encyclopedia of Genes and Genomes (KEGG) and gene ontology (GO) analyses using the R package “clusterProfiler” (Appendix A).

### 2.7. Estimation of Immune Infiltration and Immune-Related Pathway Activity

To investigate immune infiltration between the high- and low-risk groups, we computed the stromal score, immune score, and ESTIMATE score for each BRCA patient by the ESTIMATE algorithm. Moreover, the infiltration scores of 16 immune cells and the activity of 13 immune-related pathways (Appendix A) between 2 risk groups were assessed by the single sample gene set enrichment analysis (ssGSEA) in R package “GSVA”.

### 2.8. Drug Sensitivity Analysis

The sensitivities of common chemotherapeutic agents were analyzed using the R package. This R package (https://github.com/paulgeeleher/pRRophetic (accessed on 13 November 2022)), based on pre-treatment gene expression and drug sensitivity data from cancer cell lines, can project the half-maximal inhibitory concentration (IC50) of chemotherapeutic drugs in two risk groups, predicting chemotherapy response.

### 2.9. Statistics

All statistical analyses were conducted through the R software (version 4.2.1). The *t*-test (data required normality and equal variance) or Wilcoxon-test (non-parametric test) was performed to compare the means between the two groups. The Chi-square test was used to compare categorical variables. Prognostic survival curves were plotted through the Kaplan–Meier method and tested by log-rank test. The correlation analysis was performed using the Pearson correlation (based on bivariate normality) or Spearman correlation (not met parameter test requirement). LASSO regression was used to obtain the prognostic model’s characteristic coefficients and crucial genes. Univariate and multivariate Cox regression models were performed to evaluate the independent risk factors linked to survival. In all analyses, *p* < 0.05 was deemed statistically significant. The workflow chart for this study is shown in Figure 1.

## 3. Results

### 3.1. Identification of DEGs with Both EMT and DDR Functions in BRCA

To obtain DEDGs (genes with both DDR and EMT functions), we took an intersection for EMT-related genes and DDR-related genes (Figure 2A). Meanwhile, 9538 DEGs between BRCA and normal samples were identified. Subsequently, we took an intersection for DEDGs and 9538 DEGs to obtain 23 genes (DEDGs) (Figure 2B). A total of 16 of them (*AURKA*, *FOXM1*, *CCNA2*, *TACC3*, *E2F1*, *CDK5*, *TYMS*, *PTPA*, *CDKN2A*, *FOXK2*, *TRIM28*, *YWHAZ*, *BRCA1*, *CCND1*, *PIN1*, and *SKP2*) were upregulated, while 7 of them (*VIM*, *TP63*, *CCND2*, *YWHAG*, *ATM*, *HIPK2*, and *JUNB*) were downregulated in BRCA. The expression levels of 23 DEDGs are shown in Figure 2C. In addition, we analyzed the protein–protein interactions (PPI) for 23 DEDGs (Figure 2D), where 0.4 (medium confidence) was set as the interaction scoring criterion to explore the relevance of DEDGs. *CCND1*, *BRCA1*, *ATM*, *CDKN2A*, *CCNA2*, *FOXM1*, *AURKA*, *E2F1*, *CCND2*, and *SKP2* were the hub genes based on the PPI network (Figure 2E).

### 3.2. TCGA-BRCA Classification Based on the DEDGs

We conducted the consensus clustering analysis to further investigate the relationship between the 23 DEDGs and BRCA subtypes. The highest intragroup correlation was observed at k = 2 (the clustering variable (k) was increased from 2 to 5). Based on the clustering results, 23 DEDGs could divide 1050 TCGA-BRCA patients into 2 clusters (Figure 3A). The overall survival rate of patients in Cluster 1 was lower than in Cluster 2 (*p* < 0.01, Figure 3B). Furthermore, the heatmap indicated a significant difference in 23 DEDGs expression levels, age, and T-stage between the 2 clusters (*p* < 0.05) (Figure 3C).

### 3.3. Establishment of DEDGs Prognostic Model Based on the TCGA Cohort

First, the univariate COX regression was used to screen DEDGs affecting the prognosis in TCGA-BRCA patients (Figure 4A). A total of 6 prognosis DEDGs (*p* < 0.05) were selected into the LASSO regression model for further narrowing down the candidate genes, and a 6-DEDGs (TP63, YWHAZ, BRCA1, CCND2, YWHAG, and HIPK2) signature was finally established based on the best λ value (Figure 4B, C). The risk score was calculated as follows: risk score = (−0.154 × *TP63* exp.) + (0.973 × *YWHAZ* exp.) + (0.219 × *BRCA1* exp.) + (−0.227 × *CCND2* exp.) + (0.416 × *YWHAG* exp.) + (−0.471 × *HIPK2* exp.). Based on the median risk score, TCGA-BRCA patients were divided into the high-risk group (n = 525) and the low-risk group (n = 525) (Figure 4D). Principal component analysis (PCA) revealed a significant difference between the two risk groups (Figure 4E). In addition, survival analysis demonstrated that patients in the high-risk group had poorer survival than the low-risk group (Figure 4F,G). Finally, we constructed the receiver operator characteristic (ROC) curves for 5, 7, and 10 years to evaluate the model’s predictive capacity. The areas under the ROC curve (AUCs) at 5, 7, and 10 years were 0.701, 0.639, and 0.616, respectively (Figure 4H). To further validate 6-DEDGs’s rationality, we conducted survival analysis and ROC curves for 112 patients with triple-negative BRCA. The results demonstrated that high-risk patients had lower survival rates in comparison to low-risk patients (*p* < 0.05), and it had good predictive effectiveness at 5, 7 and 10 years (AUC = 0.736, 0.684 and 0.601) (Appendix A). In addition, with the prevention of losing key prognostic genes, we also included all 23 DEDGs into LASSO Cox regression to generate a prognostic model, and the final prognostic genes obtained were essentially identical (TP63, YWHAZ, BRCA1, CCND2, and HIPK2) (Appendix A).

### 3.4. Validation of the 6-DEDGs Signature Prognostic Model

To further validate the reliability of the prognostic model, 2 independent GEO-BRCA cohorts, GSE20685 (n = 327) and GSE88770 (n = 117), were used to validate the predictive model. Patients of two GEO cohorts were divided into the high- and low-risk groups using the same median risk score in the TCGA cohort (Figure 5A,F). Principal component analysis (PCA) showed substantial differences between the two risk groups of validation cohorts (Figure 5B,G). Consistent with the results of the TCGA cohort, in the GEO cohorts, patients in the high-risk group had poorer overall survival than the low-risk group (Figure 5C,D,H,I). Eventually, the prognostic model based on 6-DEDGs also had the excellent predictive capability in the two validation cohorts. The AUCs of GSE20685 at 5, 7, and 10 years were 0.700, 0.640, and 0.621 (Figure 5E), and the AUCs of GSE88770 at 5, 7, and 10 years were 0.736, 0.684, and 0.601 (Figure 5J).

### 3.5. Evaluation of the Independent Prognostic Value of the Prognosis Model

To illuminate whether the risk score based on the prognosis model was an independent prognostic factor for BRCA patients, the risk score and clinical characteristics (age, sex, TNM, and stage) were incorporated into the univariate Cox regression analysis (Figure 6A). Subsequently, variables with *p* < 0.05 (stage, age, and risk score) were included in the multivariable Cox regression analysis. As expected, the risk score was identified as an independent prognostic factor affecting the prognosis of BRCA. In addition, stage and age were also ascertained as independent prognostic factors (Figure 6B). The heatmap based on the TCGA cohort and clinicopathological information demonstrated the differential expression of DEDGs in the two groups and differences in gender, age, and survival status (*p* < 0.05) (Figure 6C). Based on these outcomes, we constructed a clinical prognostic nomogram by combining the risk score with three clinical characteristics to predict patients’ survival rates (Figure 6D). To confirm the predictive performance of the prognostic nomogram, the prognostic calibration curve was depicted and showed good predictive accuracy for survival at 5, 7, and 10 years (Figure 6E).

### 3.6. Functional Enrichment Analysis Based on the Prognosis Model

Enrichment analyses of GO and KEGG functions were conducted to explore the potential biological functions and pathways based on our risk classification. A total of 3732 DEGs were screened between the 2 risk groups. GO and KEGG enrichment results showed that DEGs were enriched intensively in various immune-related molecular functions and pathways, including cytokine–cytokine receptor interaction, PI3K-Akt signaling pathway and humoral immune response (Figure 7A–D).

### 3.7. Comparison of Immune Activity between Two Risk Groups

To obtain the potential correlation between the risk groups and immune activity, we estimated the difference in immune status between the high- and low-risk groups. Firstly, we calculated the ESTIMATE score, immune score, and stromal score for each BRCA patient based on the ESTIMATE algorithm and visualized them in a heatmap (Figure 8A). The high-risk group had lower immune scores, stromal scores, and ESTIMATE scores than the low-risk group (*p* < 0.05) (Figure 8B). Subsequently, we compared the enrichment scores of 16 immune cells and 13 immune-related pathways among different risk groups. The results indicated that the high-risk group had lower levels of immune cell infiltration compared to the low-risk group (*p* < 0.05), such as B cells, CD8+ T cells, dendritic cells (DCs), tumor-infiltrating lymphocytes (TILs), and helper T cells (T helper cells) (Figure 8C). Meanwhile, 10 immune-related pathways, including antigen-presenting cell (APC) co-inhibitory and co-stimulatory pathway, chemokine receptor (CCR), checkpoint, cytolytic activity, human leukocyte antigen (HLA), inflammation promotion, parainflammation, T cell co-stimulation, and type II interferon (IFN) response, were less active in the high-risk group (Figure 8D). According to the above results, the poor prognosis in high-risk patients may be attributed to the immunosuppressive tumor microenvironment. The results of two GEO validation sets are presented in Appendix A.

### 3.8. Drug Sensitivity Analysis between Two Risk Groups

The sensitivities of four common BRCA chemotherapy agents (Olaparib, Niraparib, Cyclophosphamide, and Talazoparib) and four other tumor chemotherapy agents (Erlotinib, Crizotinib, Axitinib, and Cisplatin) in two risk groups were analyzed using the R package “pRRophetic”. The results showed that the IC50 sensitivity scores for these drugs were significantly higher in the high-risk group (Figure 9A–H). which meant that BRCA patients in the high-risk group may be insensitive to chemotherapy treatment. The results of two GEO validation sets are shown in Appendix A. Furthermore, we predicted the drug sensitivity of the EMT-targeted drug “Sorafenib” and concluded that patients in the high-risk group were not sensitive to this drug compared to the low-risk group (Appendix A).

## 4. Discussion

With the advance in medical research, increasing biomarkers are available to predict BRCA’s prognosis, However, the BRCA prognosis predicted by single functional biomarkers may have limitations due to the complex etiology and drug resistance of BRCA [21,22]. Therefore, we developed a gene signature based on dual-function to predict prognosis, and the prognostic role of the six genes we screened has been put forward in other cancer studies [23,24,25,26,27,28]. Furthermore, this is the first study to explore novel prognostic markers for breast cancer using genes with dual functions of DDR and EMT, aiming to propose a new strategic direction for the treatment and prognostic evaluation of BRCA patients.

As mentioned previously, DDR and EMT are associated with the prognosis of BRCA patients. Therefore, we selected 23 genes with both DDR and EMT-related functions (DEDGs). Then, BRCA patients were categorized into 2 clusters by unsupervised consensus cluster analysis according to the 23 DEDGs. There were distinctions in common clinical characteristics between the two clusters, such as age and T-stage. Subsequently, a prognostic model consisting of 6-DEDGs was constructed by univariate Cox regression and LASSO Cox regression analysis. The model had better predictive capability for BRCA’s survival compared to previous prognostic genetic models (AUC = 0.5–0.6) [29,30], especially in the 5-year survival rate (AUC = 0.701). The AUC value is proportional to the model’s predictive performance, and AUC greater than 0.7 means higher predictive ability [8]. It can be seen that the prognostic model we built has a satisfactory clinical predictive power. To improve the clinical feasibility of the prognostic model, we combined the clinical features with the 6-DEDGs signature to construct a prognostic nomogram. The calibration curve demonstrated the excellent predictive survival ability of the nomogram. Our findings offer some guidance for the follow-up treatment of clinical individuals.

The prognostic model proposed in this study was made up of 6 genes (TP63, YWHAZ, BRCA1, CCND2, YWHAG, and HIPK2). Several studies reported that these six genes play an influential role in the advancement of BRCA. TP63 (TAp63 and ΔNP63 isoforms) belongs to the p53 transcription factor family, and the TAp63 subtype inhibits cancer cell metastasis, improving patient survival. Jie Mei et al. suggested that YWHAZ promotes the migration of breast cancer cells by adjusting the DAAM1/RhoA signaling pathway [31]. Emiko Hiraoka et al. demonstrated that YWHAG (also known as 14-3-3γ) promotes the migration activity of BRCA cells [32]. Tumor susceptibility gene BRCA1 is involved in transcriptional regulation, and its overexpression can increase the risk of BRCA [33]. Low expression of cell cycle protein D2 (CCND2) has been reported to promote cancer cell growth, leading to poor patient prognosis [34]. Homeodomain-interacting protein kinase 2 (HIPK2) combines with P53 to inhibit the transcriptional activity of cancer cells, blocking cancer progression [35]. To sum up, previous studies were consistent with our findings, revealing that the six genes are highly relevant to the prognosis of BRCA. In other words, our 6-DEDGs signature could somewhat predict a BRCA patient’s prognosis.

Based on the DEGs between the two risk groups, we performed GO and KEGG functional analysis and were surprised to find that many immune-related molecular functions and pathways were enriched, such as cytokine–cytokine receptor interaction, PI3K-Akt signaling pathway and humoral immune response. The immune microenvironment, playing a significant role in tumor growth and metastasis, may be an effective target for cancer therapy [36]. Therefore, we further examined the scores of immune cells and the activity of immune pathways between the two risk groups in the TCGA and GEO cohorts. The results indicated that the majority of immune cell infiltration in the high-risk group exhibited low levels, including three critical immune cells (B cells, CD8+ T cells and TILs). It was reported that B cells, derived from lymphoid progenitor cells (CLP), produce antibodies to participate in the composition of humoral immunity through the blood system. Its infiltration level is positively correlated with patients’ survival [37]. Masanori Oshi et al. indicated that in triple-negative breast cancer (TNBC), CD8+ T cells are an essential component of the tumor immune microenvironment (TIME), and their high scores are linked to a better survival rate [38]. Several studies have shown that tumor-infiltrating lymphocytes (TILs) act as an important prognostic marker for early TNBC, predicting clinical treatment and improving patient prognosis [39]. What needs to be explained is that regulatory T (Treg) cells mediate immunosuppression and are related to the poor prognosis of patients [40]. In our study, the Treg cell infiltration level was low for the high-risk group. Therefore, we speculate that the possible reason for this discrepancy is the severe impairment of immune function in advanced patients, causing the relatively low levels of all immune cells, such as Treg cells. In addition, most of the immune-related pathways in the high-risk group have lower scores, keeping consistent with the previous speculation. As a result, the prognostic model based on our six genes can forecast the immune activity of BRCA patients. Moreover, we can reasonably deduce that the poor prognosis for the high-risk group may be correlated with the immunosuppressive tumor microenvironment. In conclusion, 6-DEDGs can predict the immune infiltration level of BRCA patients.

In addition, a patient’s immune level has been proposed to be significantly related to chemotherapy’s efficacy. François et al. noted that the immune system contributes to the effectiveness of chemotherapy for patients [41]. Meanwhile, chemotherapy is the primary cancer treatment, inhibiting cancer progression by blocking the division of cancer cells [42]. Thus, we evaluated the IC50 values of 9 chemotherapeutic drugs for BRCA. The results showed that the high-risk group had higher IC50 values for the 9 anti-tumor drugs. In other words, nine anti-tumor drugs had poor treatment efficacy for high-risk group patients. Among nine anti-tumor drugs, “Olaparib” and “Niraparib” are PARP inhibitors available for DDR’s treatment, while “Sorafenib” is EMT-targeted agent. These evidences demonstrated that our prognostic model can to some extent distinguish the drug-sensitive populations. It also validated the accuracy of the prognostic model we constructed.

Research on genetic models and clinical applications are complementary, and the genetic models research can provide some guidance for clinical applications. In this study, our predictive model aimed to implement precise treatment for patients through classifying patients into high and low risk groups by gene expression. However, Gene expression in patients is achieved by expensive RNA sequencing technology [43], making it difficult to apply to clinical routine. We had analyzed the overexpression and poor prognostic value of *YWHAZ*, *BRCA1*, and *YWHAG* in BRCA, so can we reasonably speculate on developing a drug to inhibit *YWHAZ*, *BRCA1*, and *YWHAG*-related functions to suppress cancer progression and make patients have a good prognosis. For example, Serpina3n not only serves as a serine protease inhibitor, but also promotes inflammation regression [44,45]. Moreover, inflammatory dysregulation is an intrinsic mechanism in several cancers [46], and *YWHAZ* and *YWHAG* are a family of phosphoserine/threonine-binding molecules that can be indirectly degraded by Serpina3n. Therefore, we may be able to treat the population of breast cancer patients insensitive to chemotherapy with Serpina3n [47,48]. This study may provide some perspectives for BRCA’s treatment. Of course, it is a retrospective study based on data from public databases, and some information may not be available, so substantial research is still needed to validate these findings in the future.

In a nutshell, we identified 6 DEDGs with the effects on BRCA prognosis through COX regression analysis. Eventually, based on the 6 genes, we constructed the prognostic models to reasonably make predictions for the survival, drug sensitivity, and immune activity of BRCA patients. These findings in our study may serve as potential gene targets and clinical guidelines for BRCA’s treatment and prognosis evaluation.

## 5. Conclusions

The *TP63*, *YWHAZ*, *BRCA1*, *CCND2*, *YWHAG*, and *HIPK2* signature we screened could accurately forecast the prognosis, immune activity, and drug response of BRCA patients. This study provides a theoretical framework for BRCA’s treatment and contributing into individualized therapy strategies in BRCA.

## Figures and Tables

**Figure 1 ijerph-20-01221-f001:**
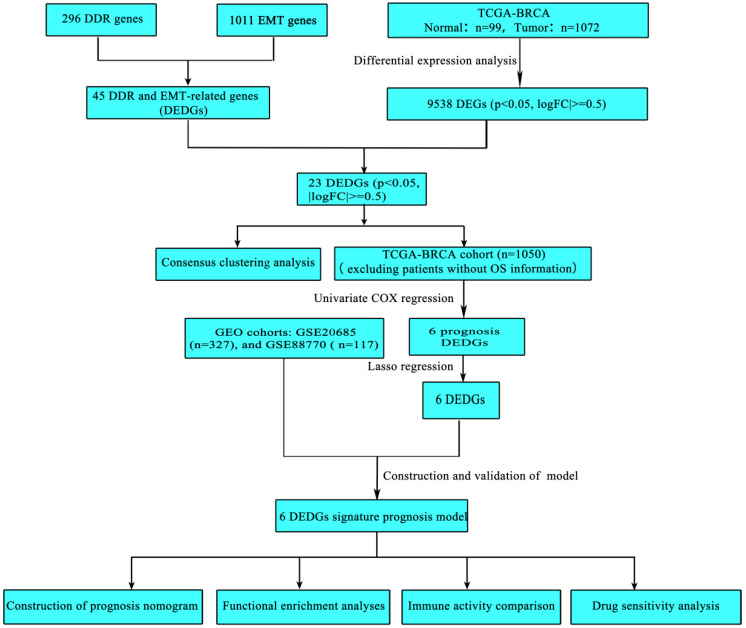
The workflow chart of this study.

**Figure 2 ijerph-20-01221-f002:**
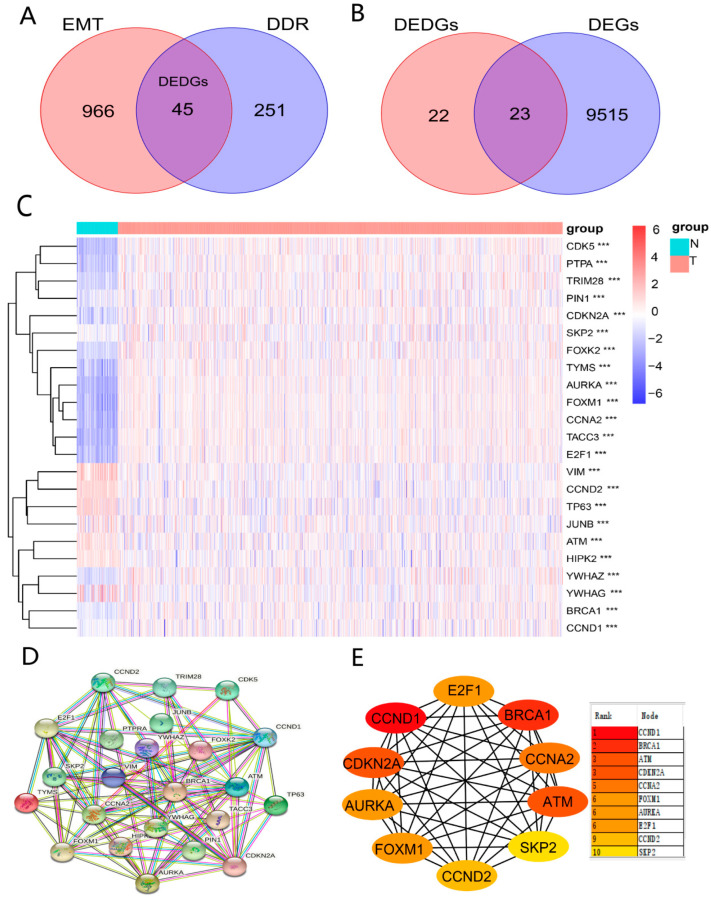
Identification of DEDGs with both EMT and DDR functions: (**A**) Venn diagram showing the intersecting genes (DEDGs) between EMT and DDR; (**B**) Venn diagram presenting the intersecting genes between DEDGs and DEGs; (**C**) heatmap of 23 DEDGs between TCGA-BRCA and normal samples (blue: low expression level; orange: high expression level; *** *p* < 0.001); (**D**) protein–protein interaction (PPI) network of 23 DEDGs (interaction score = 0.4); (**E**) 10 hub genes based on the PPI results.

**Figure 3 ijerph-20-01221-f003:**
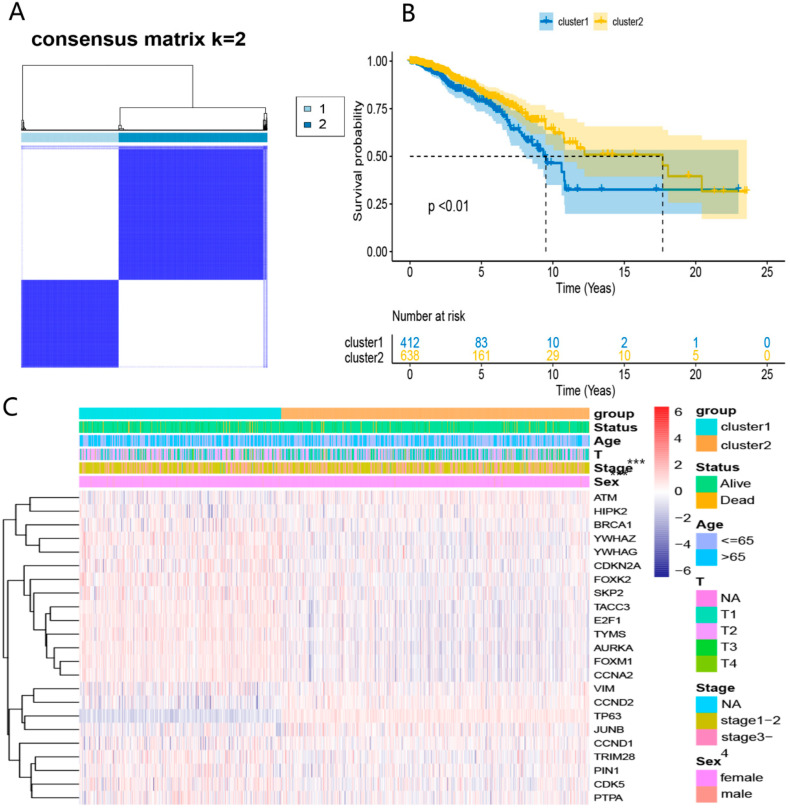
TCGA-BRCA classification based on the DEDGs: (**A**) consensus clustering at k = 2; (**B**) Kaplan–Meier OS curves for 2 clusters; (**C**) heatmap for DEDGs expression according to 2 clusters and clinical features. *** *p* < 0.05.

**Figure 4 ijerph-20-01221-f004:**
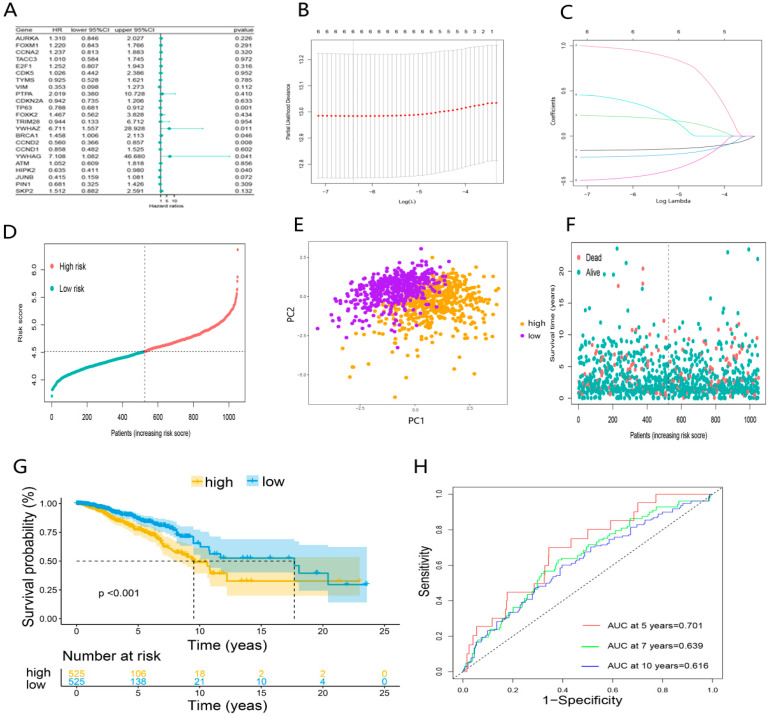
Establishment of DEDGs prognostic model based on the TCGA cohort: (**A**) forest plot of the univariate Cox regression for 23 DEDGs; (**B**) cross-validation for optimal parameter selection in the LASSO regression; (**C**) LASSO regression for 6 OS-related DEDGs; (**D**) distribution of patients based on the risk score; (**E**) principal component analysis (PCA) of 6 DEDGs; (**F**,**G**) the survival status and overall survival analysis of the two risk groups; (**H**) the time-dependent receiver operating characteristic (ROC) curve of 6 DEDGs.

**Figure 5 ijerph-20-01221-f005:**
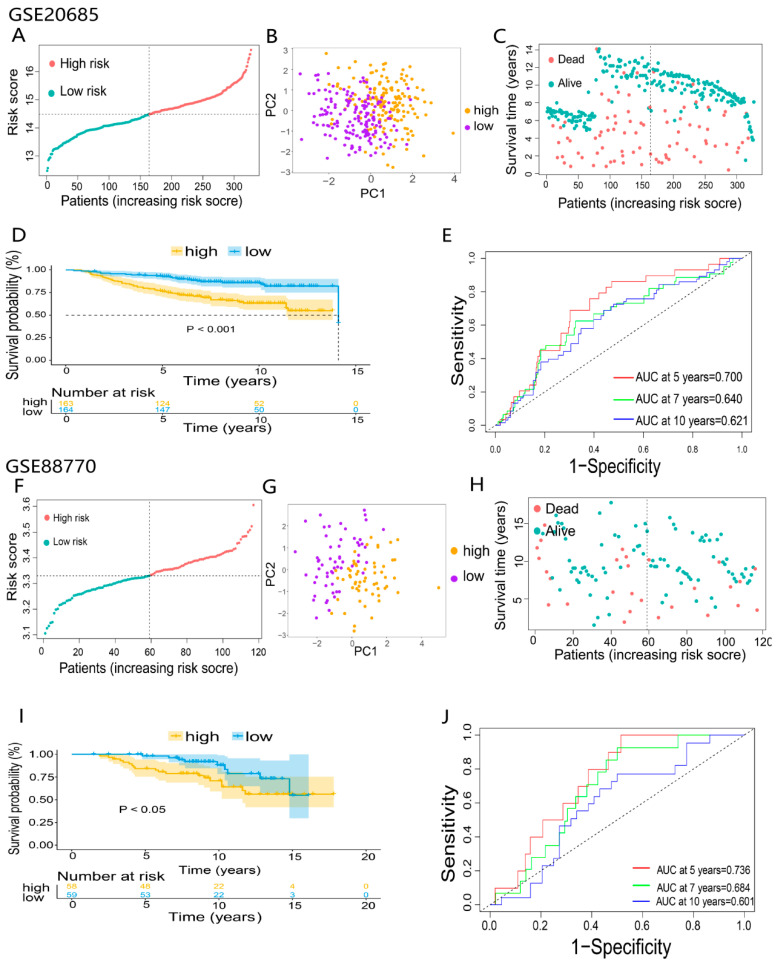
Validation of 6-DEDGs feature prognostic models: (**A**,**F**) classification of patients in the GEO validation cohorts based on the median risk score of the TCGA cohort; (**B**,**C**,**G,H**) principal component analysis (PCA) of 2 GEO validation cohorts; (**D**,**I**) the overall survival and survival status analysis of 2 GEO validation cohorts; (**E**,**J**) the time-dependent receiver operating characteristic (ROC) of 2 GEO validation cohorts.

**Figure 6 ijerph-20-01221-f006:**
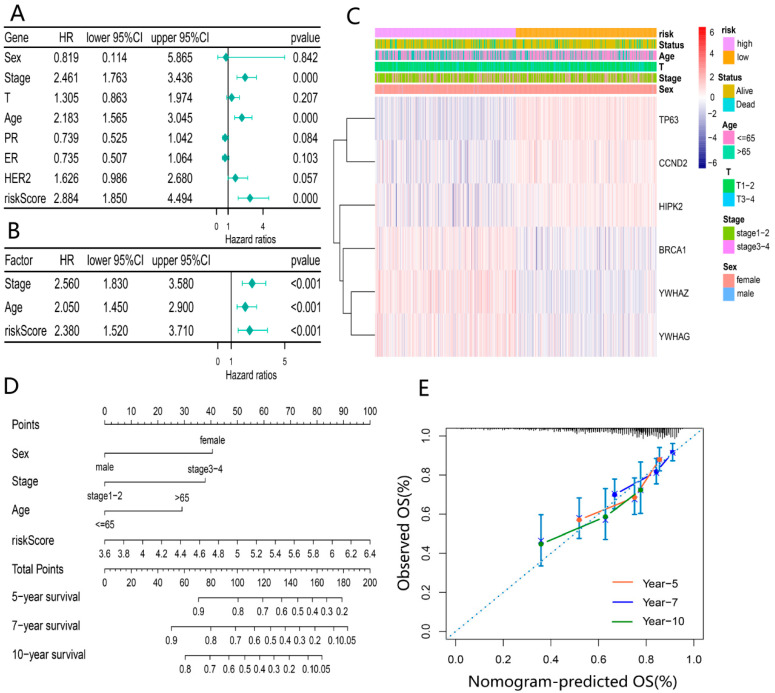
Evaluation of independent prognostic value of the risk model: (**A**) univariate COX regression analysis based on the risk score and clinical characteristics; (**B**) multivariate COX regression analysis for independent prognostic factors; (**C**) heatmap of 6-DEDGs based on different risk groups with pathological characteristics; (**D**) prognostic column line graph for predicting 5-year, 7-year and 10-year overall survival in BRCA patients; (**E**) the calibration curves of the nomogram for 5-year, 7-year and 10-year survival for BRCA patients.

**Figure 7 ijerph-20-01221-f007:**
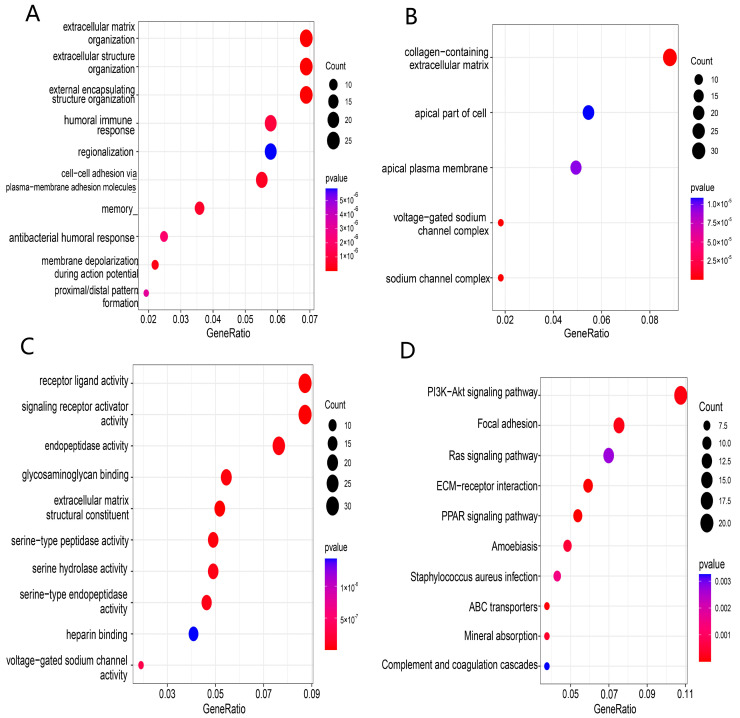
Functional enrichment analysis of TCGA-BRCA based on the risk model: (**A**–**C**) biological process (BP), cellular component (CC), and molecular function (MF) enrichment for the risk model; (**D**) KEGG pathway enrichment in the risk model.

**Figure 8 ijerph-20-01221-f008:**
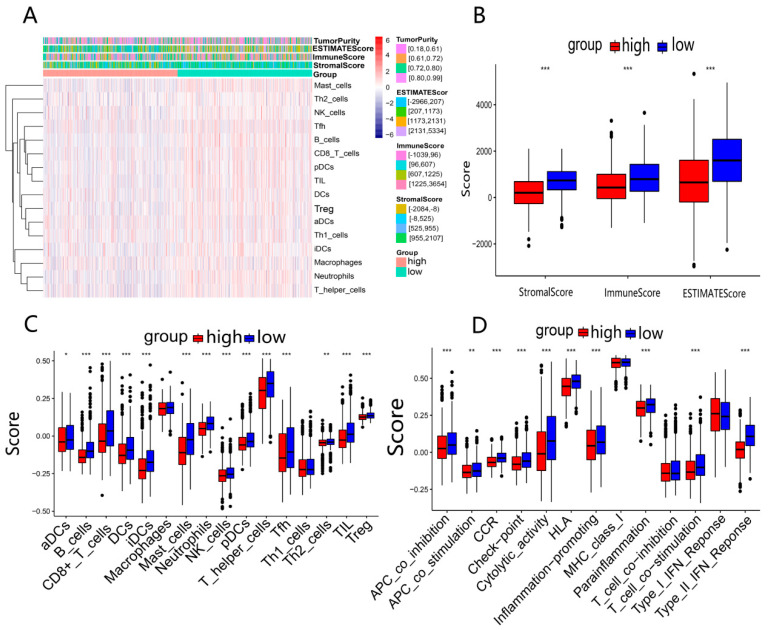
Analysis of immune activity in 2 risk groups in TCGA-BRCA: (**A**) comparison of enrichment levels of immune cells and types between 2 risk groups; (**B**) box plots used to compare the immune score, stromal score, and ESTIMATE score for 2 risk groups; (**C**,**D**) comparison of enrichment scores for 16 immune cells and 13 immune-related pathways between 2 risk groups. * *p* < 0.05; ** *p* < 0.01; *** *p* < 0.001.

**Figure 9 ijerph-20-01221-f009:**
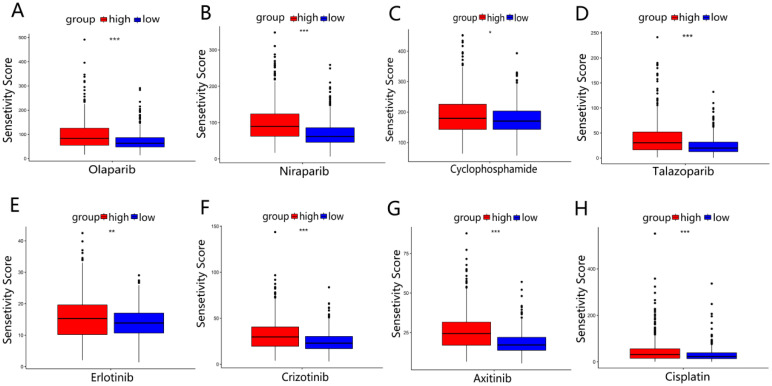
Drug sensitivity analysis in TCGA-BRCA: (**A**–**D**) the sensitivity analysis of 4 BRCA common chemotherapy agents (Olaparib, Niraparib, Cyclophosphamide, and Talazoparib) in 2 risk groups. (**E**–**H**) The sensitivity analysis of 4 other common chemotherapy agents (Erlotinib, Crizotinib, Axitinib, and Cisplatin) for 2 risk groups. * *p* < 0.05; ** *p* < 0.01; *** *p* < 0.001.

## Data Availability

The data analyzed or generated during this study are included in this article and its Appendix A.

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
