# Peer review of "Identification of a Novel Gene Signature with DDR and EMT Difunctionalities for Predicting Prognosis, Immune Activity, and Drug Response in Breast Cancer"

_ijerph, 2023, doi:10.3390/ijerph20021221_

Round 1
Reviewer 1 Report
Dear authors,
Your manuscript, "Identification of a novel gene signature with DDR and EMT functionalities for predicting prognosis, immune activity, and drug response in breast cancer", proposes a 6-gene panel to evaluate the prognosis of breast cancer (BC) patients. One of the strengths of the study focuses on the analysis of dual-function genes, which can increase the contribution of this manuscript to their field. However, I would like to comment on some concerns:
Major comments:
1. The study evaluates differentially expressed genes between tumor and control TCGA samples for then selects only genes implicated in DNA damage repair (DDR) and Epithelial-Mesenchymal Transition (EMT) pathways. However, you aim to find a panel useful for evaluating prognosis in BC patients (instead of diagnosing it). Would it be useful to start this analysis by including all dual-function differentially expressed genes? Eventually, you can add more genes to evaluate their combination.
2. TCGA data is obtained by RNA-seq while GEO data used in your study was produced using microarrays. How your score faces different technologies? The formula/model/nomogram would be the same for different quantification technologies? How do you deal with the normalization process?
3. How do you imagine this panel being applied to the clinical routine? As you reported, we can use microarray or RNAseq to evaluate the expression levels of the six genes. Nevertheless, can we translate it into affordable technologies? I suggest adding a discussion about it.
4. There are several single and panel biomarkers for BC prognosis. Have you compared the features of your proposal with others previously described?
Minor comments
5. Typo in the x-axis label of figures 3B and 4G: "Time (years)"
Author Response
Response to Reviewer 1 Comments
- The study evaluates differentially expressed genes between tumor and control TCGA samples for then selects only genes implicated in DNA damage repair (DDR) and Epithelial-Mesenchymal Transition (EMT) pathways. However, you aim to find a panel useful for evaluating prognosis in BC patients (instead of diagnosing it). Would it be useful to start this analysis by including all dual-function differentially expressed genes? Eventually, you can add more genes to evaluate their combination.
Response : Thanks for your kind advice. The certain function-specific genetic features has been widely used to explore cancer’s prognostic biomarkers. (doi: 10.1093/bib/bbac291、10.1093/bib/bbac291 and 10.3389/fimmu.2022.843515). It may be more effective to focus some function to screen out key prognostic markers. Additionally, EMT and DDR are critical factors in cancer progression and patient poor prognosis (doi: 10.1016/j.critrevonc.2017.04.005、10.3892/or.2013.2341 and 10.4161/cc.6.19.4754), and there are certain associations between them (doi: 10.1038/ncb3013 and 10.1016/j.molcel.2016.08.009). Consequently, we would like to identify crucial genes with both EMT and DDR functions, and then use statistical models to make reasonable predictions about the prognosis and treatment of cancer patients. In addition, your suggestions are enlightening. So, we further performed LASSO Cox regression analysis including 23 all differentially expressed genes with EMT and DDR dual functions. The final results showed that 5 genes were strongly associated with prognosis in breast cancer patients (TP63, YWHAZ, BRCA1, CCND2 and HIPK2), and we proceeded to analyse survival analysis and ROC curves. The predictive effect was consistent with our previous results (Figure S2 in Supplementary file 9), which proved the genes we screened may be reasonable. This section description had been added to the revised manuscript.
- TCGA data is obtained by RNA-seq while GEO data used in your study was produced using microarrays. How your score faces different technologies? The formula/model/nomogram would be the same for different quantification technologies? How do you deal with the normalization process?
Response : RNA-seq and microarrays are completely different techniques, the normalization methods of the two data are also different. Therefore, we did not directly combine the data from different sources for comparison, but we normalized them separately. TCGA provides the normalized TPM (Transcripts Per Kilobase of exon model per Million mapped reads) data, and TPM was transformed by log2 (data + 1) as the gene’s expression level in our article. The partial GEO data has been normalized, so before analysing the data, we used the R package "Boxplot" to check whether the data is in normalized state. If the data were not normalized, we would normalize the data with the function "normalizeBetweenArrays" in the R package "limma".
Our model gives signature gene a relative weight coefficients and thus can be applied to normalized gene expression data. In addition, we used the same cutoff value (50%) based on model to divide the risk population in GEO and TCGA, and used the ROC and calibration curves to validate the effect in different data from the different sources. The results both indicated that our formula/model/nomogram can be well applied in BRCA patients of TCGA and GEO.
- How do you imagine this panel being applied to the clinical routine? As you reported, we can use microarray or RNAseq to evaluate the expression levels of the six genes. Nevertheless, can we translate it into affordable technologies? I suggest adding a discussion about it.
Response : Thanks for your practical suggestion. Genetic modelling research and clinical applications are inseparable. Regarding this section, I had added a discussion section into the manuscript.
- There are several single and panel biomarkers for BC prognosis. Have you compared the features of your proposal with others previously described?
Response : Thanks for your kind suggestion. The model's predictive performance was assessed by the AUC value, and AUC greater than 0.7 means higher predictive ability (doi: 10.1186/s12931-022-02110-w). The predictive ability of our screened prognostic genetic markers was higher (AUC=0.7) compared to previous prognostic markers for BRCA (AUC=0.5-0.6) (doi:10.1155/2021/2649123 and 10.1186/s12859-022-04894-6). Therefore, the prognostic markers we screened for were relatively more favourable. This section had been added to the discussion section in the manuscript.
- Typo in the x-axis label of figures 3B and 4G: "Time (years)"
Respose : Thank you for your attentive advice. Two images had been modified.

Reviewer 2 Report
In this manuscript Zhang and colleagues have used Bioinformatic tool and claiming for the identification of novel gene signature with DDR and EMT.
In my view claim made by authors are fully misleading as neither in vitro nor in vivo data presented in this study. The gene signature predicted in this study must be validated in lab by using specific cell line or tumor models.
Transcriptional data and dug sensitivity data is essential to attract readers.
Author Response
Response to Reviewer 2 Comments
In my view claim made by authors are fully misleading as neither in vitro nor in vivo data presented in this study. The gene signature predicted in this study must be validated in lab by using specific cell line or tumor models.
Transcriptional data and dug sensitivity data is essential to attract readers.
Response : Thanks for your kind suggestion. The data excavation and experimental validation are not separable, and the data excavation could provide some direction for the experiments. This similarly kinds of work also had been published by several group (doi : 10.1093/bib/bbac291、10.3389/fimmu.2022.855849、10.1186/s12967-020-02323-x、10.1186/s12967-021-02952-w and 10.1186/s12967-019-02173-2). This study was a retrospective study based on public database data and aimed to uncover critical prognostic markers with dual functions of EMT and DDR to provide some views on the BRCA's prognosis and treatment. Of course, we believe that extensive experiments are needed in the future to further validate these findings.

Reviewer 3 Report
In this paper, Zang at al explore the dual-function of EMT- and DDR-related genes in BRCA prognosis. The authors identified a 6-genes signature showing impact on BRCA prognosis and overall survival, immune involvement and predicted chemotherapy efficiency. However there are few points that should be clarified to produces a complete and useful study.
The functional enrichment analysis is not clear who the immune-related functions and pathways where selected. Might be worthy to include (as supplementary data) the list of genes contribution for each function/pathway and respective q-value. Since the GeneRatio is low, the number of genes DEG in the function/pathways is low (some of the highlighted pathways only has 1 gene). For example, other pathways as PI3K-Akt signaling has a bigger contribution. To control the FDR, it would be recommended to adjust the p-values when is possible (e.g. functional enrichment analysis).
The drug sensitivity analysis, it was selected 4 drugs for BRCA (3 PARP inhibitor + 1 alkylating agent/immune inhibitor) and 4 drugs for treatment of other solid tumours which mostly targets the cell proliferation. Since the main focus of this paper was EMT and DDR, should be interesting to see additional drugs which impact on ETM and/or DDR (e.g. Icaritin).
Since BRCA has different subtypes, is this model aplicable with the same sensitivity and specificity to different subtypes?
The TCGA-BRCA database is increasing, currently with 1098 cases. The access date of the database should be mentioned in the methods, data collection.
Author Response
Response to Reviewer 3 Comments
- The functional enrichment analysis is not clear who the immune-related functions and pathways where selected. Might be worthy to include (as supplementary data) the list of genes contribution for each function/pathway and respective q-value. Since the GeneRatio is low, the number of genes DEG in the function/pathways is low (some of the highlighted pathways only has 1 gene). For example, other pathways as PI3K-Akt signaling has a bigger contribution. To control the FDR, it would be recommended to adjust the p-values when is possible (e.g. functional enrichment analysis).
Response : Thanks for your thoughtful advice. Tables for GO and KEGG analysis had been added to the supplementary file 6 and 7. Your suggestion was very intelligent and practical, we lowered the P-value to 0.01, making the GO and KEGG analysis more accurate and the GeneRatio improved accordingly. The corresponding texts and images in this section had been corrected in the manuscript.
- The drug sensitivity analysis, it was selected 4 drugs for BRCA (3 PARP inhibitor + 1 alkylating agent/immune inhibitor) and 4 drugs for treatment of other solid tumours which mostly targets the cell proliferation. Since the main focus of this paper was EMT and DDR, should be interesting to see additional drugs which impact on ETM and/or DDR (e.g. Icaritin).
Response : Thanks for your enlightening advice. Incorrect DDR is one of the important factors promoting cancer progression, and PARP inhibitor as a target drug for DDR, which has been acknowledged by most research (doi: 10.1002/1878-0261.13224、10.1186/s13046-021-02005-6). In this study, "Olaparib" and "Niraparib" are PARP inhibitors available for DDR's treatment. In addition, with your inspiration, we proceeded to predict the drug sensitivity of Icaritin for BRCA in TCGA and GEO database. Unfortunately, since the R package "Oncopredict" we used was not able to detect the Icaritin's IC50, we cannot predict the drug's sensitivity. However, we predicted the drug sensitivity of the EMT-targeted drug "Sorafenib" and concluded that patients in the high-risk group were not sensitive to this drug compared to the low-risk group. (Figure S3 in Supplementary file 9). This section was consistent with what we expected and had been added to the article manuscript.
- Since BRCA has different subtypes, is this model aplicable with the same sensitivity and specificity to different subtypes?
Response : Thank you for your instructive suggestions. Thus, we screened 112 patients with triple-negative breast cancer from TCGA database for survival prognosis analysis with 6 DEDGs. The results showed that survival rates were lower among high-risk patients compared to low-risk patients (P< 0.05). Moreover, we predicted AUC values for 5, 7 and 10 years with 0.736, 0.684 and 0.601 respectively, suggesting a relatively good predictive efficiency. It also proved that the model we constructed had good reasonability. The section had been added in the revised manuscript.
- The TCGA-BRCA database is increasing, currently with 1098 cases. The access date of the database should be mentioned in the methods, data collection.
Response : Thanks for your great suggestions. The access dates of data collection had been added in the revisd manuscript.

Round 2
Reviewer 1 Report
Dear authors,
Your manuscript, "Identification of a novel gene signature with DDR and EMT functionalities for predicting prognosis, immune activity, and drug response in breast cancer", proposes a 6-gene panel to evaluate the prognosis of breast cancer (BC) patients. One of the strengths of the study focuses on the analysis of dual-function genes, which can increase the contribution of this manuscript to their field. I thank you for having answered my previous concerns. Once I believe the manuscript was improved successfully, I endorse this submission.
Author Response
Response to Reviewer 1 Comments
- Your manuscript, "Identification of a novel gene signature with DDR and EMT functionalities for predicting prognosis, immune activity, and drug response in breast cancer", proposes a 6-gene panel to evaluate the prognosis of breast cancer (BC) patients. One of the strengths of the study focuses on the analysis of dual-function genes, which can increase the contribution of this manuscript to their field. I thank you for having answered my previous concerns. Once I believe the manuscript was improved successfully, I endorse this submission.
Response : Thank you for your kind reply and recognition of this work. Also, we appreciate greatly for your contribution to enhancing the article quality.

Reviewer 2 Report
Authors didn't work on my suggestion. I would like to see experimental validation.
Author Response
Response to Reviewer 2 Comments
Authors didn't work on my suggestion. I would like to see experimental validation.
Response : Thanks for your suggestions. The current work was aimed to uncover critical prognostic markers with dual functions of EMT and DDR, providing some views on the BRCA's prognosis and treatment. And this similarly kinds of work also had been published by several group (doi : 10.1093/bib/bbac291、10.3389/fimmu.2022.855849、10.1186/s12967-020-02323-x、10.1186/s12967-021-02952-w and 10.1186/s12967-019-02173-2). On the other hand, we also believe that experimental validation is more effective. We may perform the experiments to validate the findings combinded other analysis in the future.
